# Global environmental implications of atmospheric methane removal through chlorine-mediated chemistry-climate interactions

Atmospheric methane is both a potent greenhouse gas and photochemically active, with approximately equal anthropogenic and natural sources. The addition of chlorine to the atmosphere has been proposed to mitigate global warming through methane reduction by increasing its chemical loss. However, the potential environmental impacts of such climate mitigation remain unexplored. Here, sensitivity studies are conducted to evaluate the possible effects of increasing reactive chlorine emissions on the methane budget, atmospheric composition and radiative forcing. Because of non-linear chemistry, in order to achieve a reduction in methane burden (instead of an increase), the chlorine atom burden needs to be a minimum of three times the estimated present-day burden. If the methane removal target is set to 20%, 45%, or 70% less global methane by 2050 compared to the levels in the Representative Concentration Pathway 8.5 scenario (RCP8.5), our modeling results suggest that additional chlorine fluxes of 630, 1250, and 1880 Tg Cl/ year, respectively, are needed. The results show that increasing chlorine emissions also induces significant changes in other important climate forcers. Remarkably, the tropospheric ozone decrease is large enough that the magnitude of radiative forcing decrease is similar to that of methane. Adding 630, 1250, and 1880 Tg Cl/year to the RCP8.5 scenario, chosen to have the most consistent current-day trends of methane, will decrease the surface temperature by 0.2, 0.4, and 0.6 °C by 2050, respectively. The quantity and method in which the chlorine is added, its interactions with climate pathways, and the potential environmental impacts on air quality and ocean acidity, must be carefully considered before any action is taken.

The Paris agreement in 2015 states that greenhouse gas (GHG) emissions should be reduced so that the anthropogenic global temperature increase is limited to 2 °C above preindustrial levels while pursuing efforts to limit the increase even further to 1.5 °C[1]. Recent reports show that GHG emission and concentration trends are not consistent with

this target[2,3]. About 1.2 W m$^{-2}$ of the total 2.7 W m$^{-2}$ increase in radiative forcing from 1750 to 2019 is due to the direct and indirect effects of methane (CH$_4$), whose concentration has increased by ~150% during the Anthropocene[4]. CH$_4$ has both anthropogenic and natural origins which are similar in magnitude[5] thus making emission control a

✉ e-mail: mahowald@cornell.edu; a.saiz@csic.es

difficult task. $CH_4$ is mostly removed from the atmosphere through the chemical reaction with the hydroxyl radical (OH; Eq. 1) and to a lesser extent through the reaction with the photolytic chlorine atom (Cl; Eq. 2). The further photochemical reaction of $CH_3O_2$ results in the production of two important GHGs, tropospheric ozone ($O_3$) and carbon dioxide ($CO_2$) (chemical scheme–Fig. S1). $CH_4$ is transported by tropical upwelling to the stratosphere where it produces significant concentrations of water vapor, another GHG (Eq. 1)[6,7].

$$CH_4 + OH \xrightarrow{O_2} CH_3O_2 + H_2O \qquad (1)$$

$$CH_4 + Cl \xrightarrow{O_2} CH_3O_2 + HCl \qquad (2)$$

$CH_4$ has a much shorter lifetime in the atmosphere (10–12 years) compared to other potent GHGs (several tens to hundreds of years)[4,8]. Meeting the 1.5 °C temperature goal requires substantial cuts in the emissions and atmospheric burden of $CO_2$[9] and $CH_4$. Some studies proposed intentionally adding chlorine to the atmosphere to decrease $CH_4$ concentration through iron salt aerosol[10,11]. Horowitz, et al.[12] have shown that an increase in global methane occurs as the result of unintentional chlorine additions. Their main focus was to study marine cloud brightening as a means of reducing radiative forcing through the generation of additional sea-salt aerosol (SSA). However, all these studies (including intended and unintended chlorine additions) did not quantify the amount of chlorine that is needed to achieve a reduction in global methane.

For the current atmospheric conditions, reactive chlorine species originate from heterogenous reactions on sea-salt aerosols and other tropospheric aerosols, coal- and biomass-burning, and industrial activities[13–15]. Chlorine-containing species undergo photochemical reactions that produce chlorine atoms; the chlorine atom provides a direct sink towards $CH_4$ (Eq. 2) and depletes $O_3$ (Eq. 3; a critical source of tropospheric OH), therefore increasing $CH_4$ loss via (Eq. 2) (Cl) but reducing $CH_4$ loss via Eq. (1) (OH) (further details in Supplementary Text S1). A recent study showed that for present-day halogen abundance, chlorine together with iodine and bromine chemistry in the atmosphere decreases global $CH_4$ loss, thus increasing $CH_4$ lifetime, concentration, and radiative forcing[16]. However, the potential impacts of a significantly larger atmospheric chlorine burden on atmospheric composition, radiative forcing, and surface temperature remain unexplored.

$$Cl + O_3 \rightarrow ClO + O_2 \qquad (3)$$

Here, we apply a well-documented Earth system model (Community Earth System Model, CESM version 1.1) coupled with comprehensive halogen sources and chemistry[16] to explore for the first time the potential influence of adding chlorine emissions. We adopt the Representative Concentration Pathway 8.5 (RCP8.5) as the main baseline scenario because it provides the closest representation of global $CH_4$ burden for the present-day conditions[17], about 5200 Tg for the year 2022[18]. We conduct twelve sensitivity simulations from the present-day (2020) to mid-century (2050) in which molecular chlorine ($Cl_2$) is emitted at a constant rate over all ocean surfaces, with emission fluxes between 10 and 1880 Tg Cl/year (henceforth designated as S10 to S1880 scenarios). We also conduct two additional sensitivity cases with RCP6.0 as the baseline. We summarize the simulation cases in Table S1.

This paper quantifies the globally averaged impact of additional chlorine emissions as a potential climate intervention technique. A homogeneous addition of chlorine species over all ocean surfaces may not be feasible in this respect but is chosen here as a pragmatic starting point. We consider a multitude of cases with different chlorine emissions but omit regional analysis to show a synthesis of the global impacts on atmospheric chemistry and climate. This includes

analyzing the global change in atmospheric composition, both the intended change to $CH_4$ and the unintended changes to other atmospheric constituents (mainly to OH, tropospheric $O_3$, sulfate aerosol, stratospheric $O_3$, and stratospheric water vapor), and determining the associated radiative forcing and surface temperature response to these changes. Additionally, we indicate the possible environmental impacts due to the addition of chlorine, including the impacts on air quality and ocean acidification. We identify several uncertainties in our modeling results. Finally, we propose an agenda for future research on this potential climate mitigation methodology.

## Results and discussion

### Nonlinear response of methane burden and lifetime to chlorine emissions

The atmospheric response to the additional chlorine emissions (Methods) is highly complex and nonlinear (Fig. 1). Based on our CESM modeling results, adding 90 Tg Cl/year (S90; Table S1) can be regarded as an important threshold for the response of $CH_4$ to chlorine changes. This is comparable to tripling the current-day chlorine atom burden (Fig. S2). Below this threshold, increases in tropospheric chlorine emissions (S10 scenario) from RCP8.5 first lead to an increase in the atmospheric $CH_4$ burden compared to RCP8.5 in 2030; the emission scenarios of S40, S60, and S80 result in approximately the same $CH_4$ burden and lifetime as S10 (Fig. 1 inset). Such an increase in methane lifetime is due to an increase in global chlorine burden as was previously shown by Horowitz et al.[12]. Here we note that above this emission threshold of 90 Tg Cl/year, the global $CH_4$ burden begins to decrease.

This nonlinear response of $CH_4$ burden to the increase in chlorine emissions is explained by the change in the overall $CH_4$ lifetime (Fig. 1) induced by changes in tropospheric photochemistry involving $O_3$, OH, and other species[12]. The additional tropospheric chlorine atoms act to consume tropospheric $O_3$ through atmospheric chemical reactions initiated by Eq. (3). Supplementary figure S3 shows the percentage of ocean area (~70%, ~50%, ~20%, ~15%, and ~10% in RCP8.5, S10, S630, S1250, and S1880 cases, respectively) that has a higher reactivity of Cl toward $O_3$ (Eq. 3) compared to $CH_4$ (Eq. 2). This shows that above the addition of 10 Tg Cl/year, most of the $O_3$ above ocean is consumed and the reactivity of Cl shifts towards the intended reaction with $CH_4$ (Eq. 2). Given that tropospheric $O_3$ is the primary source of the tropospheric oxidant, OH[19], the reduction in $O_3$, caused by the additional chlorine flux, results in a reduction in the OH burden (Fig. S4b), which in turn controls $CH_4$ oxidation through Eq. (1). Therefore, below the 90 Tg Cl/year threshold, the additional chlorine decreases tropospheric $O_3$ and OH, thereby leading to a decrease in the $CH_4$ loss by OH, without being fully compensated for by a sufficiently large $CH_4$ loss by the Cl atom itself. In contrast, above this chlorine emission threshold, the loss of $CH_4$ by reaction with Cl atom becomes large enough to offset the $CH_4$ loss by OH resulting from $O_3$ decreases (see SI text for more detailed discussion).

To better understand the $CH_4$ response to the addition of chlorine to the atmosphere above this threshold, we analyze in detail the baseline case (RCP8.5) with current-day chlorine atom burden (i.e., 0.7 Mg global chlorine atom burden, Fig. S2), and four additional scenarios, S10, S630, S1250, and S1880. The S10 scenario results in an increase in $CH_4$ burden, while the scenarios S630, S1250, and S1880 result in a removal of 20%, 45 and 70% of $CH_4$ atmospheric burden compared to RCP8.5 by 2050 (Fig. 2). When an equivalent Cl emission is added to a different baseline scenario (RCP6.0), the reduction in $CH_4$ lifetime and the burden is different. $CH_4$ burden is reduced by 2300 and 2200 Tg by the year 2030 in the RCP8.5 S1250 and RCP6.0 S1250, respectively, while $CH_4$ lifetime is reduced by 5.5 and 4.5 years by the year 2030 in the RCP8.5 S1250 and RCP6.0 S1250, respectively (Fig. S7, S8, and S9). This highlights that the atmospheric response to additional chlorine emissions is sensitive to the global burdens of methane, $O_3$, and other critical atmospheric species.

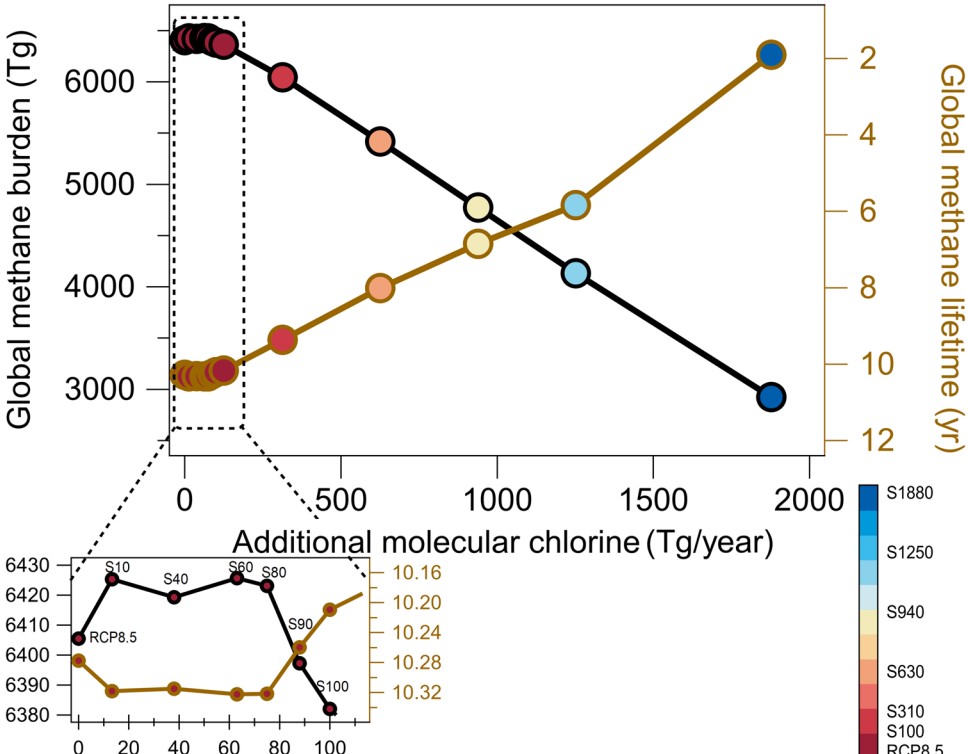

**Fig. 1 | The relationship between additional molecular chlorine emissions, global CH₄ burden (black line; left axis), and the CH₄ e-folding chemical lifetime (brown line; right axis–reversed).** The CH₄ burden is shown for the year 2030, while the 10-year average (2020–2030) is shown for CH₄ lifetime. The CH₄ lifetime is defined as the chemical lifetime of CH₄ (obtained via dividing the atmospheric CH₄ burden by the CH₄ chemical loss rate). The colormap shows the scenarios (detailed setup listed in Table S1). Inset plot–results from RCP8.5, S10, S40, S60, S80, S90, and S100 scenarios highlighting the nonlinear response of global CH₄ (burden and lifetime) to additional molecular chlorine flux. As molecular chlorine emissions are increased from an additional 10 Tg Cl/year (S10) up to an additional 80 Tg Cl/year (S80), there is a slight increase in the global CH₄ burden. Only by increasing emissions above 90 Tg Cl/year (S90) does the global CH₄ burden and its lifetime decrease.

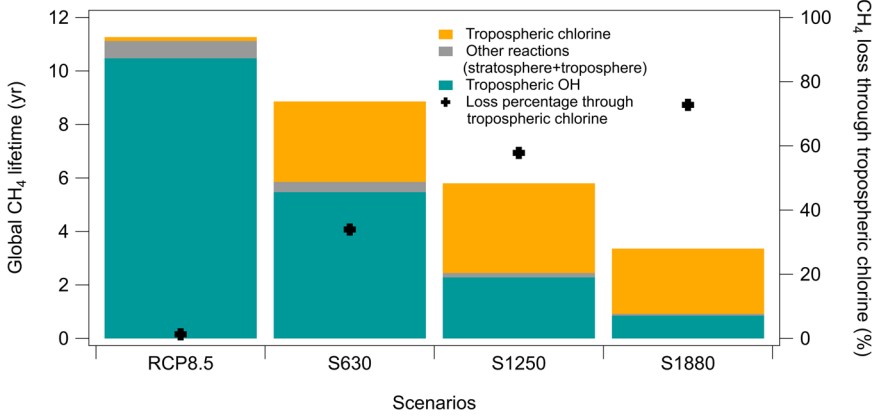

**Fig. 2 | The e-folding chemical lifetime of methane in the atmosphere to chemical loss in the year 2050 in RCP8.5 and for scenarios S630, S1250, and S1880 (left axis).** Note that this is a different time period (2050) than in Fig. 1 (2030). The relative contribution of tropospheric OH, tropospheric chlorine, and other chemical loss pathways to the global integrated CH₄ lifetime is represented by the fraction of the bar from each source. The crosses indicate the fraction of the loss of CH₄ from tropospheric chlorine (right axis). More than 630 Tg/year of additional molecular chlorine emissions are needed to make tropospheric chlorine the dominant loss pathway of CH₄. The temporal variations of each CH₄ loss and lifetime channel are shown in Fig. S5 and S6.

The chlorine atom becomes an important chemical loss pathway for methane (Eq. 2) in the scenarios where chlorine emissions are increased above the threshold of 90 Tg Cl/year (Fig. 2). In the RCP8.5 baseline case, in 2050 the chlorine sink represents <2% of the CH₄ loss. However, as chlorine emissions are increased, the CH₄ loss through chlorine atoms increases (Fig. 2) thus decreasing the lifetime and burden of CH₄ (Fig. S4, S5, and S6). In 2050, chlorine accounts for about 30% of the CH₄ loss in scenario S630. An emission increase of 1250 Tg Cl/year is needed to reduce the methane lifetime by >50% (Fig. 2). This occurs when the Cl-driven methane destruction (Eq. 2) takes over the otherwise dominant OH-driven CH₄ losses.

Here we have analyzed only two future emission pathways (RCP8.5 and RCP6.0) out of the many possibilities. As we have shown

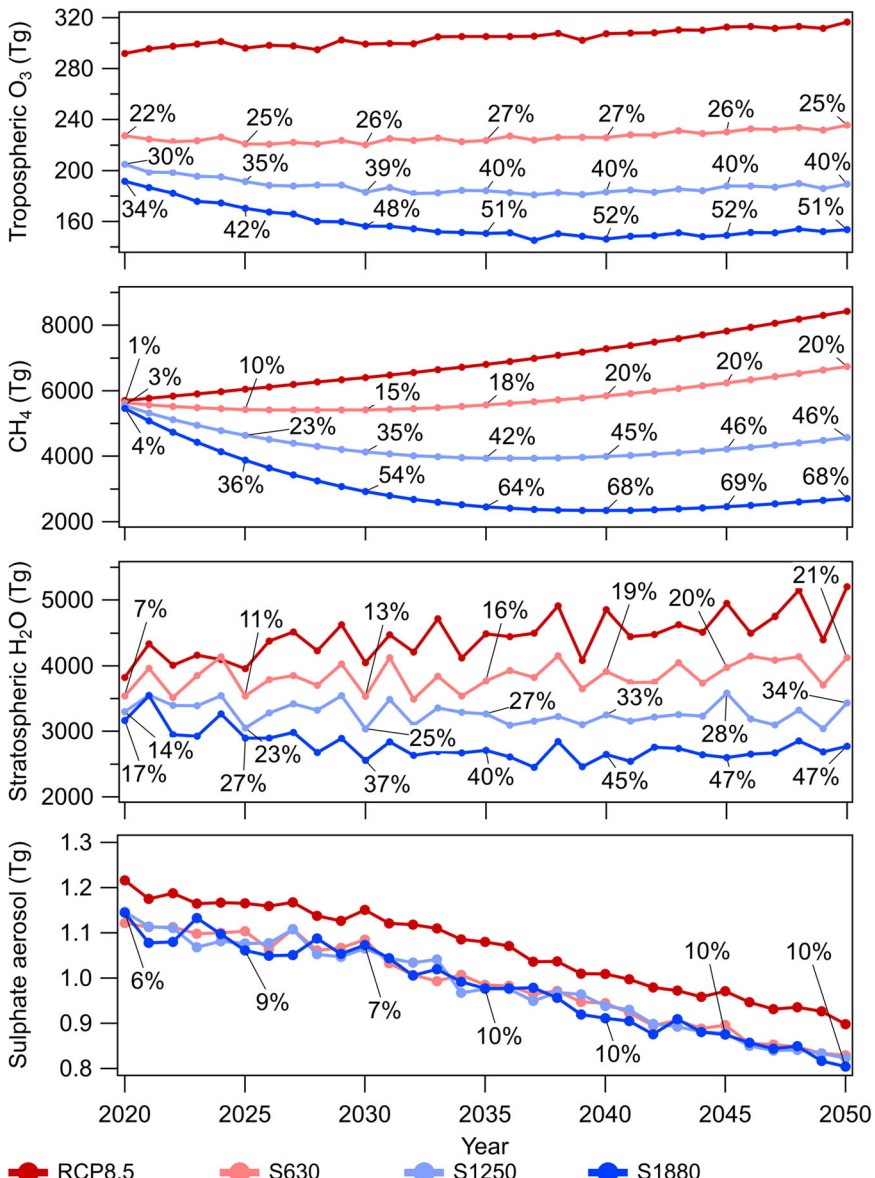

**Fig. 3 | Global burdens of short-lived climate forcers from 2020 to 2050 in different scenarios (red, RCP8.5; pink: S630; light blue: S1250; dark blue: S1880).** Top to bottom panels—tropospheric $O_3$, $CH_4$, Stratospheric $H_2O$, and sulfate aerosol. The percentage reduction compared to RCP8.5 is shown as numbers on each plot. The global burdens of all SLCFs are lower compared to RCP8.5. The percentage change stabilizes after 2035 for all SLCFs in all the cases.

above, the methane response to added chlorine has a threshold above which additional chlorine reduces methane lifetime and concentrations (Fig. 1). The concentrations of $CH_4$ and $O_3$ in the base scenarios will affect this threshold. Similarly, scenarios with more locally concentrated emissions of Cl may also change this threshold due to the nonlinear chemistry involved.

## Impacts on short-lived climate forcers

The addition of chlorine modifies the global integrated burden of key short-lived climate forcers (SLCF; $CH_4$, tropospheric $O_3$, stratospheric $H_2O$, and sulfate aerosol) (Fig. 3). The atmospheric $CH_4$ burden is reduced by ~20%, ~45%, and ~70% by year 2050 in the S630, S1250, and S1880 scenarios, respectively (Fig. 3). By the year 2050, the additional chlorine emissions also lead to lower tropospheric $O_3$ by ~25%, ~40%, and ~51%, and a reduction in stratospheric $H_2O$ by ~21%, ~34%, and ~47%, respectively, for the three mitigation scenarios, as compared to RCP8.5. Lastly, the change in tropospheric OH results in decreased secondary sulfate aerosol production, mainly as a result of less $SO_2$

conversion into $H_2SO_4$. Sulfate aerosol decreases by about 10% for all the mitigation scenarios compared to the RCP8.5 by mid-century. Other secondary aerosol types, such as secondary organic aerosol, change negligibly due to compensating effects between increased Cl atom and reduced OH. It is noteworthy that the chlorine-mediated relative change to all SLCFs stabilizes after about 15 years (the late 2030s) suggesting that the atmospheric system stabilizes at a new steady state after the chlorine additions (Fig. 3).

Increasing the loss of $CH_4$ through Cl will decrease the $CH_4$ burden for about a decade (the 2020s in Fig. 3). However, $CH_4$ increases again at the end of the simulation period (the 2040s) because the $CH_4$ emissions are still increasing in RCP8.5. The decrease in the $CH_4$ burden in scenario RCP6.0 S1250 slows down by the end of the simulation period (Fig. S9b).

## Impacts on surface temperature

Both $CH_4$ and tropospheric $O_3$ increase in RCP8.5 from the present-day to the mid-21st century. These increases contribute to global warming

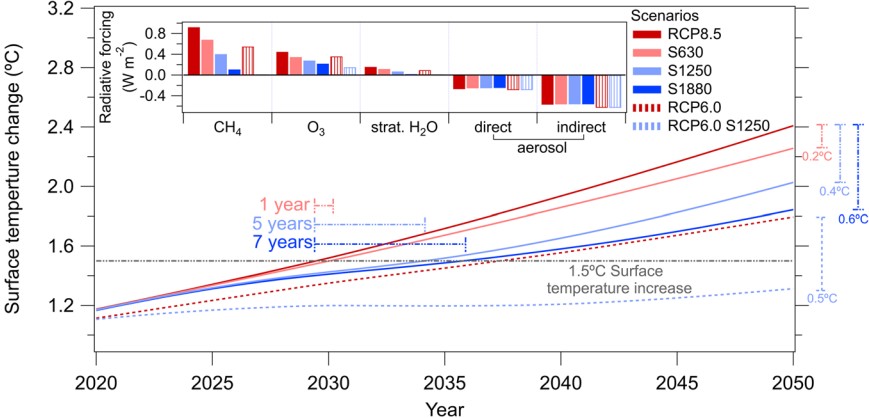

**Fig. 4 | Surface temperature change from 2020 to 2050 for the RCP8.5, S630, S1250, and S1880 (solid lines) and RCP6.0 and RCP6.0 S1250 (dashed lines) scenarios.** The 1.5 °C target is marked by the horizontal gray dash-dotted line. The "gained" time to cross the 1.5 °C target compared to RCP8.5 is shown in the colored horizontal dash-dotted lines for each scenario. The 2050 temperature decreases for each scenario compared to RCP8.5 is shown in the colored vertical dash-dotted lines on the right-hand side. The S1250 scenario adds 5 years before reaching the 1.5 °C target compared to the RCP8.5 scenario. The S1250 scenario results in a reduction of 0.4 °C in 2050 from the surface temperature given in RCP8.5. Additional emissions of 1250 Tg Cl/year to RCP6.0 will postpone crossing the 1.5 °C target until after 2050. The inset shows the radiative forcing in 2050 w.r.t 1850 of the five main SLCFs that are impacted by the added molecular chlorine emissions. Radiative forcing of CH₄ decreases by almost 90% from RCP8.5 to the S1880 scenario. O₃ and stratospheric H₂O are both affected by the increase in additional molecular chlorine flux and their radiative forcing decrease as a result. The impact of changes in atmospheric aerosols is small.

and threaten the 1.5 °C target set by the UNFCCC[1,2]. Using a simple energy model, the Model for the Assessment of Greenhouse gas Induced Climate Change (MAGICC), the impact on radiative forcing and temperatures induced by the changes in atmospheric composition upon adding chlorine emissions is calculated (Methods) and shown in Fig. 4. While the focus here is on RCP8.5 with its higher CH₄ emissions, we also show the impacts of added $Cl_2$ on a scenario with lower GHG emissions (RCP6.0).

Compared to RCP8.5, our results show that the removal of 45% atmospheric CH₄ in 2050 (S1250) decreases the radiative forcing of CH₄ and tropospheric O₃ by 55% and 40%, respectively, leading to a temperature reduction of 0.4 °C. Adding the same chlorine emission to RCP6.0 as the baseline case, i.e., the RCP6.0 S1250 case, the radiative forcing of tropospheric O₃ is cut in half and that of CH₄ and stratospheric H₂O by 2050 is reduced to insignificant values (Fig. 4 inset). Interestingly, the unintended response to this mitigation scenario, the reduction in radiative forcing of tropospheric O₃ and stratospheric H₂O, is comparable to the response from the removal of CH₄. Emission additions of 630 and 1880 Tg Cl/year reduce the surface temperature in 2050 by 0.2 °C and 0.6 °C, respectively, in the S630 and S1880 scenarios. The surface temperature response in the RCP6.0 S1250 scenario decreases the 2050 temperature by 0.5 °C compared to the RCP6.0 baseline case. The aerosol direct and indirect radiative forcing in all mitigation scenarios (those in both RCP8.5 and RCP6.0) is slightly reduced by about 1% compared to their respective baseline.

These scenarios demonstrate that the rate of increasing surface temperature can in principle be slowed down significantly by increasing chlorine emissions and reducing methane. This results in the postponement of the point where the 1.5 °C warming target is exceeded (Fig. 4). In RCP8.5 scenario, the 1.5 °C target is expected to be exceeded in the 2020s; adding 630, 1250, and 1880 Tg Cl/year postpones crossing the 1.5 °C target by ~1 year, 5 years and 7 years, respectively. RCP6.0 is projected to cross the 1.5 °C target by the late 2030s; in the RCP6.0 S1250 scenario, temperatures do not cross the 1.5 °C target even at mid-century, highlighting that the atmospheric implications of increasing chlorine emissions are linked to the evolution of GHGs affecting global warming.

## Potential environmental impacts

The mitigation scenarios explored here represent a significant anthropogenic source of chlorine atoms to the atmosphere (i.e., 630, 1250, and 1880 Tg Cl/year additional chlorine flux), with the highest flux reaching that of natural sea salts (3000 Tg Cl/year)[20]. This requires careful consideration of the associated environmental impacts.

Although molecular chlorine is a known toxin at levels greater than 34 ppbv[21], globally averaged surface $Cl_2$ mixing ratios reach up to 2 ppbv in the S1880 scenario (Fig. S11c). Secondly, globally averaged surface mixing ratios of O₃, an extensively monitored hazardous atmospheric gas with various harmful effects on human health and vegetation[22–24], are expected to decrease by 50, 70, and 85% compared to RCP8.5 in the S630, S1250, and S1880 scenarios, respectively (Fig. S11a); however, the possible regional impact of increased chlorine levels on O₃ needs to be carefully assessed. The addition of chlorine does not substantially change the global average of surface CO (Fig. S12) and NOₓ (Fig. S11b), or the NO to NO₂ ratio (Fig. S13; a measure of atmospheric oxidative capacity). However, regional values may change and need to be explored in future studies.

The modeled deposition of chemical acidity to the surface ocean in the form of HCl from Eq. (2) is potentially an order of magnitude larger than the acidity flux caused currently by anthropogenic reactive sulfur and nitrogen. Assuming that the majority of the injected $Cl_2$ is eventually removed as HCl deposition into the ocean, the three model scenarios result in upper bounds of 18, 35, and 53 Tmol Cl/year for RCP8.5 S630, S1250, and S1880, respectively. This is substantially larger than the estimated anthropogenic global ocean deposition of reactive sulfur (0.8 Tmol S/year) and reactive nitrogen (2.7 Tmol N/year) from fossil fuel combustion and agriculture[25]. While not as large as the acidification caused by ocean uptake of anthropogenic $CO_2$, reactive sulfur and nitrogen fluxes are thought to exacerbate regional acidification in coastal waters downwind of anthropogenic sources. The effects of Cl emissions on surface ocean acid-base chemistry will depend on the specifics of the $Cl_2$ injection process. For example, an injection method that uses iron to release $Cl_2$ through a catalytic cycle[26] would impact HCl in the atmosphere differently from emitting $Cl_2$ directly. However, the possibility of a separation in the NaOH and HCl could still generate substantial changes to surface ocean acid-base chemistry[27].

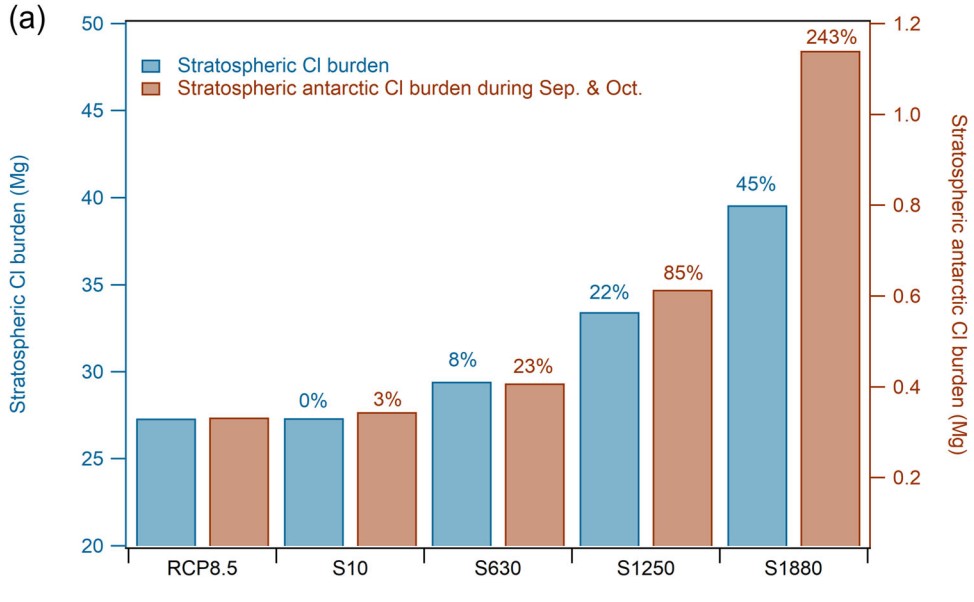

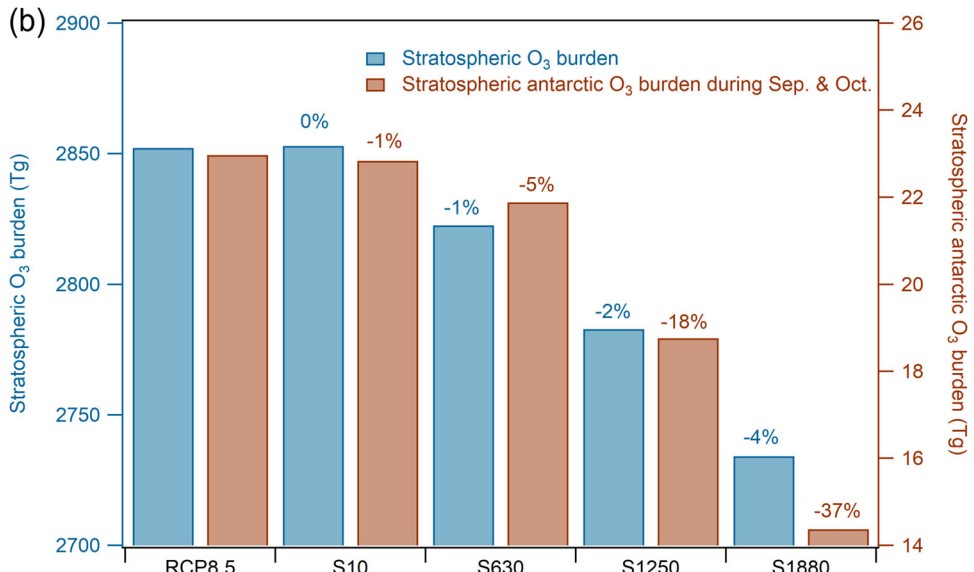

**Fig. 5 | Stratospheric chlorine and O₃ burden average from 2020 to 2050 in RCP8.5, S10, S630, S1250, and S1880 scenarios. a** Cl burden in the global stratosphere and that in the Antarctic stratosphere (90°S to 70°S; 70 hPa to 200 hPa) during September and October. **b** O₃ burden in the global stratosphere and that in the Antarctic stratosphere (90°S to 70°S; 70 hPa to 200 hPa) during September and October. The numbers on top of the sensitivity scenarios bars indicate the changes of chlorine/O₃ in these scenarios compared to that in RCP8.5 case. Note that both Y-axes in both panels start at values larger than zero.

The enhancement in tropospheric chlorine atom also increases the stratosphere burden of chlorine (as much as a 44.9% increase in S1880 scenario averaged from 2020 to 2050) and with even more in the Antarctic region (up to 242.6% increase; Fig. 5a). The significant increase in stratospheric chlorine results in enhanced depletion of stratospheric O₃ (Eq. 3) (1.0, 2.4, and 4.1% lower stratospheric O₃ in S630, S1250, and S1880, respectively, compared to RCP8.5, averaged from 2020 to 2050; Fig. 5b) which counter-acts the current stratospheric ozone recovery due to the phase-out of long-lived CFCs, HCFCs, and Halons following the implementation of the Montreal Protocol. This is most important for the future evolution of the Antarctic O₃ hole during September and October, where we found 4.7, 18.3, and 37.4% lower stratospheric O₃ in S630, S1250, and S1880, respectively, compared to RCP8.5 (Fig. 5b).

### Environmental policy implication

Results from this work focus on the global and annual averages, facilitating an initial assessment of the implications of adding chlorine on global burdens of SLCFs and their associated radiative forcing, but does not evaluate consequences for human wellbeing, because these are strongly linked to local impacts important for air quality and crop productivity. Still, some of the indicated global changes are large enough to cause concern, pointing to a need for further research of local impacts before considering adding chlorine to the atmosphere. From a climate change mitigation standpoint, the results show that chlorine addition can offer a possibility to avoid exceeding global warming of 1.5 °C, but only if combined with ambitious efforts to reduce anthropogenic net emissions of CH₄ and CO₂ in the next decades. Hence, the possibility of atmospheric Cl addition becomes a question of weighing the severe risks of climate change impacts

against the possibly severe negative environmental impacts associated with Cl addition. While many of the risks in crossing the 1.5 °C target are embedded in public knowledge, adding large amounts of chlorine to the atmosphere will have significant risks of its own. With these risks being global in scope, the possibility of this approach calls for an urgent need for an international legal framework to manage the risks.

## Uncertainty analysis

In this study, we have examined the potential impact of the proposed approach on a global scale. However, it should be noted that the response to this approach will vary regionally depending on the concentrations of ozone, $NO_x$, and other pollutants. Despite chlorine emissions being confined to the ocean surface, chlorine may still reach inland areas, resulting in different environmental implications compared to marine environments. Although these effects are accounted for in the model, they may be masked by the use of global averages.

It is important to note that the pH of aerosols is not specifically calculated in CESM. The effect on the pH of aerosols of any of the proposed methods of increasing the reactive chlorine burden in the atmosphere, whether through directly emitting reactive chlorine, adding a substance that activates chlorine in aerosols, producing new SSA from the ocean, or any other method, is not clear. Pye et al.[28] review the current state of atmospheric acidity and found that even drastic changes in sulfur dioxide and nitrogen oxide emissions across the US and Canada did not have a proportional effect on aerosol pH; on the other hand, clouds and fog exhibit a higher sensitivity to such changes. Further investigation is necessary for a more accurate analysis of the acidity change resulting from additional chlorine emissions.

The method used to inject chlorine will also affect the radiative forcing (RF). For example, if an additional substance is used to react with the chlorine in the aerosol and release it to the gas phase, that substance will modify the optical properties of the aerosol resulting in a different RF response and a change in the surface temperature response.

Our modeling results show the tipping point of the $CH_4$ response to the addition of chlorine for the RCP8.5-based scenarios occurs at 90 Tg Cl/year. However, this threshold will largely depend on $O_3$ and $CH_4$ concentrations and other atmospheric conditions. Therefore, the threshold should not be considered an absolute number but a general tipping point that has to be calculated for specific atmospheric conditions and chlorine injection methods.

Overall, these factors should be explored in future studies to better understand the potential impact of this approach on a regional scale and refine our understanding of its effectiveness to allow policy makers to conduct a cost-effectiveness analysis before any implementation is considered.

## Future research agenda

Sensitivity studies conducted here show that as we inject chlorine into the atmosphere uniformly over the ocean, the chemistry responds in a nonlinear manner. With RCP8.5 as the baseline, our modeling results suggest that 90 Tg Cl/year additional molecular chlorine flux is needed in order to obtain a decrease in $CH_4$ burdens (lower additions lead to an increase in methane burden). Tropospheric $O_3$, OH, sulfate aerosols, and stratospheric $H_2O$ decrease with additional Cl, leading to changes in global radiative forcing and lowering surface temperatures. While the original goal of the approach was to reduce methane, the unintentional outcome, especially the reduced $O_3$, resulted in additional reductions in radiative forcing that are comparable to that of methane itself. As proposed elsewhere[8,29], mitigation of methane by Cl addition would slow down the increase in global temperatures, however, as shown in this study, at the risk of possibly severe environmental impacts.

Many questions need to be addressed in future studies to better understand the impacts of adding chlorine to reduce methane. We identify the following research questions as the most important to focus on: (1) How will future emission pathways change the impact of added chlorine? (2) What are the impacts of increased chlorine emissions on air quality? (3) What are the long-term ecosystem effects of additional chlorine emissions on acid deposition over land and ocean? (4) What is the environmental footprint (e.g., energy cost, $CO_2$ emissions, etc.) of the production, transportation, and deployment stage necessary to increase the atmospheric chlorine burden? (5) Multi-model intercomparison studies should be conducted to investigate the impacts of various increased tropospheric chlorine burdens on atmospheric composition, climate, and the Earth system. (6) If the positive environmental effects overweight the negative environmental effects, how could we generate, transport, and release the quantity of chlorine studied here? (7) Where, how much, and when should chlorine emissions occur for maximum impact on climate and minimum environmental damage? An important part of future studies will be considerations of environmental justice.

## Methods
### CESM model simulation
The Community Atmospheric Model with Chemistry, version 4.0 (CAM-Chem), within the Community Earth System Model, version 1.1 framework (CESM), has been used for this study[30]. The current configuration uses a version of CAM-Chem with improved representations of very short-lived halogens (Cl, Br, and I). Further descriptions of the halogen mechanism implemented in CAM-Chem can be found in the following references:[31–34]. CAM-Chem includes all of the physical parameterizations of CAM4[35].

All the simulations were performed with a horizontal resolution of 1.9° latitude by 2.5° longitude and 26 vertical levels, from the surface up to ~3 hPa (~40 km). The emission and lower-boundary condition of air pollutants and long-lived GHGs (except for $CH_4$) follow the standard RCP8.5 scenario[36]. We used emission inventories of $CH_4$ following Li et al.[16]. The sources of naturally emitted reactive halogen species are calculated online following Iglesias-Suarez et al.[37]. The lower-boundary conditions and emissions of anthropogenic chlorine species as well as the chlorine activation and recycling processes on sea-salt aerosols are based on Keene et al.[38], Hossaini et al.[15], Claxton et al.[39], and Li et al.[16]. The model simulations are conducted in free-running mode to enable the feedback of atmospheric composition changes to the climate and vice versa.

We first conducted a 60-year spin-up (1960 to 2020) to ensure a stabilized atmospheric $CH_4$ burden. From 2020 onward, we conducted a series of sensitivity cases with various emissions scenarios of additional molecular chlorine from the ocean surface worldwide. The emission flux of molecular chlorine is constant (in the unit of molecule/$m^2$/s, therefore favoring total emissions in the tropical regions) on overall oceanic surfaces and during the entire simulation period (starting from 2020), without imposing any diurnal cycle. In this conceptual study, we do not link our modeling setup to any specific climate intervention technique method (e.g., via spraying iron salts or marine cloud brightening via sea-salt aerosol injection). Instead, we adopt a simple model setup to emit $Cl_2$ over the global oceanic surface and quantify the global impacts of the increased chlorine burden on atmospheric composition and climate systems. We have taken the following considerations into account when assuming the additional chlorine is emitted over the ocean surface, instead of in the free troposphere or over land: (1) to allow a feasible emission method that does not require aircraft; (2) to reduce the energy cost and associated $CO_2$ emissions required to emit chlorine; (3) to make full use of sea-salt aerosol, a natural chloride-containing atmospheric species prevalent in the marine boundary layer; (4) to reduce the potentially harmful

effects on humans (over land); (5) to reduce the injected amount of chlorine to the stratosphere.

Table S1 in the supplement shows the setup of the standard simulations and sensitivity cases. The names of the sensitivity cases are defined as the added molecular chlorine flux. The difference in various species between RCP8.5 and the sensitivity cases represent the impact of these additional chlorine sources on atmospheric composition. Two more scenarios under the RCP6.0 case were also added to show a possible range of the additional molecular chlorine impacts under different climate scenarios.

## MAGICC simulations

We simulate the change in surface temperature relative to 1850–1900 that results from the RCP8.5 and RCP6.0 cases and mitigation scenarios using a reduced-complexity model MAGICC version 6[40,41]. This simple climate model is divided into four boxes that are used to represent the land and ocean in the northern hemisphere, and southern hemisphere. Using the RCP8.5 and RCP6.0 initial concentrations given in MAGICC, we drive the energy-balance component of MAGICC6 with the timeseries of surface $CH_4$, sulfate aerosols, and $O_3$ computed globally and simulate the change in surface temperature and radiative forcing for 1850 through 2050 for all future cases and scenarios. Our aim with this methodology is to show the potential influence that these mitigation scenarios have on surface temperature. Therefore, the change in surface temperature and radiative forcing could best be defined as the global surface temperature change relative to 1850 from possible future addition of molecular chlorine fluxes.

## Data availability

The CESM data generated in this study have been deposited in the Mendeley Data. https://doi.org/10.17632/md85gzkmg9.1.

## Code availability

The software code for the CESM model is available from http://www.cesm.ucar.edu/models/.

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

## Acknowledgements
Q.L., C.A.C., and A.S.-L. are funded by the European Research Council Executive Agency under the European Union's Horizon 2020 Research and Innovation Program (Project ERC-2016- COG 726349 CLIMAHAL to A.S.-L.). D.M., N.M.M., and P.H. would like to acknowledge the support of Silverlining and Spark Climate Solutions. RPF would like to thank the financial support from ANPCyT (PICT 2019–2187). J.A.A. is supported by a grant from the Spanish State Research Agency (Project ODEON—TED2021-132172B-I00 to J.A.A.) and his work at IQFR-CSIC is supported by a grant from the Spanish Ministry of Universities and the Universidade de Vigo. The EPhysLab is funded by the Xunta de Galicia under Grant ED431C2021/44. The CESM project is supported primarily by the National Science Foundation (NSF). This work is supported by grant no. AGS-1906719 from the Atmospheric Chemistry Division of the U.S. National Science Foundation (NSF). M.S.J. and M.v.H would like to acknowledge support from Silverlining and Spark Climate Solutions. This material is based upon work supported by National Center for Atmospheric Research (NCAR), which is a major facility sponsored by NSF under the Cooperative Agreement 1852977. Computing resources, support, and data storage were provided by the Climate Simulation Laboratory at NCAR's Computational and Information Systems Laboratory (CISL), sponsored by the NSF. We thank the helpful suggestions from David Mann and Erika Reinhardt at Spark Climate Solutions.

## Author contributions
A.S.-L. and N.M.M. devised the research. Q.L. and A.S.-L. initiated the study in collaboration with D.M., P.H., and N.M.M. Q.L. and A.S.-L., with the help of C.A.C., R.P.F., D.E.K., and J.-F.L., developed and performed the CESM simulations. D.M., P.H., and N.M.M. conducted the MAGICC simulations. J.A.A. and L.H.-I. contributed to the policy implication. S.D. and M.S.J. contributed to the environmental impact analysis. M.v.H., M.S.J., and T.R. contributed to the data analysis. Q.L., D.M., P.H., N.M.M., and A.S.-L. wrote the manuscript with contributions from all co-authors.

## Competing interests
The authors declare no competing interests.

## Additional information

Qinyi Li [1,2,14,15], Daphne Meidan [3,15], Peter Hess[4], Juan A. Añel [5,1], Carlos A. Cuevas [1], Scott Doney [6], Rafael P. Fernandez[7], Maarten van Herpen[8], Lena Höglund-Isaksson [9], Matthew S. Johnson [10], Douglas E. Kinnison[11], Jean-François Lamarque [12], Thomas Röckmann[13], Natalie M. Mahowald [3] ✉ & Alfonso Saiz-Lopez [1] ✉

[1]Department of Atmospheric Chemistry and Climate, Institute of Physical Chemistry Blas Cabrera, CSIC, Madrid 28006, Spain. [2]Department of Civil and Environmental Engineering, The Hong Kong Polytechnic University, Hong Kong 999077, China. [3]Department of Earth and Atmospheric Sciences, Atkinson

Center for a Sustainable Future, Cornell University, Ithaca, NY, USA. [4]Department of Biological and Environmental Engineering, Cornell University, Ithaca, NY, USA. [5]EPhysLab, CIM-Uvigo, Universidade de Vigo, Ourense, Spain. [6]Department of Environmental Sciences, University of Virginia, Charlottesville, VA, USA. [7]Institute for Interdisciplinary Science (ICB), National Research Council (CONICET), FCEN-UNCuyo, Mendoza, Argentina. [8]Acacia Impact Innovation BV, Acacialaan 9, 5384 BB Heesch, The Netherlands. [9]Pollution Management group (PM), International Institute for Applied Systems Analysis (IIASA), 2361 Laxenburg, Austria. [10]Department of Chemistry, University of Copenhagen, Universitetsparken 5, DK–2100 Copenhagen Ø, Denmark. [11]Atmospheric Chemistry Observations & Modeling Laboratory, National Center for Atmospheric Research, Boulder, CO, USA. [12]Climate and Global Dynamics Laboratory, National Center for Atmospheric Research, Boulder, CO, USA. [13]Institute for Marine and Atmospheric Research Utrecht, Utrecht University, Princetonplein 5, 3584CC Utrecht, the Netherlands. [14]Present address: Environment Research Institute, Shandong University, Qingdao, China. [15]These authors contributed equally: Qinyi Li, Daphne Meidan. ✉e-mail: mahowald@cornell.edu; a.saiz@csic.es

