## [Peer Review File · Nature Communications]

REVIEWER COMMENTS

Reviewer #1 (Remarks to the Author):

Li et al: Enhancing atmospheric methane removal through Chlorine mediated chemistry-climate interactions.

General Comments

This is an important paper that reports some extremely interesting results. The research is very innovative and appears to be competently done with appropriate methodology (though I can't check it). The paper addresses a very important problem that is major topic of public debate. The findings will attract a great deal of interest both from the media and from policy makers. I recommend publication after some revision.

Background

There is a strong lobby, especially in the US and Europe, to 'geo-engineer' our way out of a significant part of global warming, by spraying salts into the troposphere (e.g. Oesté et al. *Earth Syst. Dynam.*, 8, 1–54, 2017). Covering up the methane problem would then allow the fossil fuel jamboree to continue unimpeded, and hence this is an idea much liked by many investors and with significant political traction, not just in the US but also engaging oil and coal producing interests in many countries

Atmospheric chemists have generally been intuitively very sceptical of these ideas, on the general principles of "If you are in a hole, stop digging" ; "the cover-up is worse than the crime"; and "There was an Old Woman who swallowed a fly/horse/dead of course" . But to date this intuitive scepticism has not really been backed up by detailed modelling to challenge the geo-engineering ideas. That absence has tended to leave the debate open to the proponents of planetary-scale action.

Findings

Li et al enter this debate by using the standard CESM 1.1 Earth system Model and throwing in a great deal of chlorine, as proposed by the geo-engineering lobby. Then Li et al have a good look at what happens. Their results are quite surprising and there are several counter-intuitive findings. Two important issues stand out.

1. Li et al show that throwing chlorine into the air does not make things better unless you throw in a really huge amount of Cl. The threshold is 130 Tg Cl/yr. That's 3000 Titanics, or the payload of 7 million C-130 Hercules flights. They show it's above this threshold that injecting the chlorine enables an important chemical loss pathway for methane. To cut the methane lifetime by 50% needs an injection of 1900 Tg.
2. Li et al also show that chlorine emissions have a major impact on ozone, and injection of so much chlorine has a double effect on global warming – it decreases the warming impact of ozone as much as the impact of methane. So there's a big benefit here. assuming it doesn't matter that the acidification will cause a large-scale rearrangement of atmospheric chemistry and indeed the chemistry of the top of the ocean where the chlorine eventually rains out.

Comments.

I am not able to check the modelling and methodology for mistakes. The CESM model is appropriate and the sensitivity simulation modelling appears to be done with standard methods. The 60 yr methane spin up seems reasonable. The text is well-written and clear, and the findings are presented in a lucid way.

I have one point of detail; the term "lifetime" is used for the e-folding lifetime (around 12 yr). That's quite normal in the modelling community (and indeed in IPCC) but it is not the 'lifetime' that is standard in the methane measurement community and in most public communications. For the non-modeller side of the methane debate, the "lifetime" is simply the geochemical lifetime - the burden divided by the flux - the replacement lifetime. That's about 9.2 years (e.g. Lan et al. 2021, *Phil Trans Roy Soc Lond A* 379: 20200440 – see their eqn 2.3. As this paper is going to be read widely outside the modelling community, I'd suggest using that version of 'lifetime'.

Now for several questions, which the authors should consider.

1. In the later sections of the paper (environmental policy) and in discussion of the global warming impact of removing the methane, the authors make no mention of the energy costs of the action

of adding billions of tonnes of chlorine. Yet they are seriously considering injecting between 950 and 2840 Tg of Cl a year – that is roughly between 1 and 3 billion tonnes of chlorine, more if it is wafted up as iron chloride. To put that into context, pre-covid there were about 40 million aircraft take-offs in 2019, or less than 20 million in 2020. Assume each plane is an 80 tonne Boeing 737, then 40 million x 80 = somewhat over 3 billion tonnes. In other words, the geoengineers are proposing to inject annually into the troposphere a mass of the iron chloride equal to the total mass of aircraft that fly annually. That mass lifting of chlorine will have an energy cost – in effect a carbon dioxide cost – equal to the whole modern airline industry's emissions. So maybe that cost needs to be discussed here.

2. Perhaps there should also be a brief mention of the CO₂ emission costs of mining, pulverizing, and transporting several billion tonnes of iron chloride.

3. The impact on tropospheric ozone is one of the most interesting findings in this paper, but perhaps there also should be a discussion of implications in the model for the Brewer-Dobson circulation. Much of the chlorine will have to be injected around the inter-tropical convergence zone, to allow the light and [OH] abundance to bite on the methane. But that inevitably means a lot of the chlorine will be injected up into the tropical stratosphere and the chloring will then travel round the Brewer-Dobson loop to the Polar Vortex...and it will do a lot of interesting things on the way.....

Conclusion

To conclude: this paper is remarkably interesting and potentially important. The paper is clearly only scratching the surface of a very big puzzle, but it's a good start. While it is unlikely the geoengineers will get their way, it's nevertheless possible they will indeed try something – it only takes one billionaire – and thus it is a problem that deserves attention. I strongly recommend the paper should be published – but after some revision.

Reviewer #2 (Remarks to the Author):

Li et al. investigated the effects of mitigating methane by emitting additional chlorine. They conducted sensitivity simulations in CESM and analyzed the influences of adding molecular chlorine on methane, ozone, their radiative forcing, and other environmental impacts. I totally agree with the author that “the potential environmental impacts on air quality and ocean acidity, must be carefully considered before any action is taken”. Climate issue is definitely important, but it's tricky that whether it is the time to need some last resort to kill methane. Considering well-known harmful impacts of chlorine such as toxicity and acidity, proposing to emit large amount of reactive chlorine, in my opinion, is not conveying very good information.

The manuscript is interesting and well written, but I didn't find the results to be sufficiently novel, clear, and important for a high impactful journal. Most important, the authors added chlorine emissions in their simulations as molecular chlorine, but didn't clarify what chemical form of chlorine they proposed to add. Maybe one of the reactive chlorine gases (Cl₂, HOCl, etc.)? Due to different chemistry and depositions, depending on what species is going to emit, the analysis and conclusions in the manuscript could be different.

Specific comments:

- p3, end of first paragraph: both here and in abstract, the authors mentioned the addition of chlorine to the atmosphere has been proposed to mitigate global warming. However, the cited refs (10-12) here are more likely to be the traditional geoengineering. They emitted salt containing chloride, which is different from emitting molecular (reactive) chlorine. Could you please list some refs who have proposed adding reactive chlorine like you did here?

- In addition, some parts of refs (10-12) did quantify the possible consequences. The related statement is not true.

- p7, 2nd paragraph: the threshold of 3.4×10^4 ppt looks too high, is this a typo? Furthermore, the toxin is related to chlorine gas (Cl₂), not Cl atom. Reactive chlorine gases have much larger

concentrations than Cl.

- p5, last paragraph and fig.3: when saying sulfate aerosol, is it mean sulfate only or sulfate-ammonium-nitrate aerosol? Won't the chlorine induced large pH change affect the partitioning of this total inorganic aerosol?

Reviewer #3 (Remarks to the Author):

This paper discusses the impact of future increases in tropospheric chlorine on methane lifetime and its climate impact. This discussion is framed in the context of intentional additional Cl emissions as a method to mitigate surface warming and covers scenarios with very large increases.

The author team are experts in chemistry-climate modelling and the effects of halogens. In this work they have used a state-of-the-art chemistry-climate model.

The paper contains some interesting model sensitivity experiments but, in my opinion, it is not suitable for publication in a form like this in a journal like Nature Communications. The feedbacks of Cl on CH₄ lifetime have already been discussed in recent studies cited, so the motivation for this paper in such a high-level forum seems exaggerated.

My main points are summarised below.

- 1) Recent papers, e.g. ref #11 and the group's own Nat Comms paper #16, have already noted the feedback on Cl emissions on the CH₄ lifetime. We know that the naïve suggestion from some geoengineering papers on scaling the current impact of Cl on CH₄ is wrong. Therefore, I do not think that the motivation is there for another Nat Comms paper.
- 2) The paper seems to fall between the aim of a high level discussion (in the main text) and providing the quantitative model detail (in the Supplementary Information). With so many references to the 16 supplementary figures it is difficult to read if one wants to understand the details and limitations of the model. I am left with the feeling that the authors should have better written a longer paper in a subject-specific journal where the model results could be presented clearly. I note that the motivating references #10, #11 and #12 are all journals that allow more detailed discussion.
- 3) There must be considerable model uncertainties in the specific numbers quoted, but this is not acknowledged. The abstract gives some numbers but the method (i.e. running a complex model) is not stated. To the non-expert I think the numbers are presented with more confidence than they can really be known (e.g. the 130 Tg/yr tipping point). Models can differ significantly in their O₃ burden, halogen burden and so on. This paper presents some results in too absolute a way.
- 4) Following 3) I think an important issue for any research agenda would be multi-model intercomparisons to test the robustness and quantitative details of the feedbacks discussed.

Response to Reviewer #1:

We thank the reviewer for the comments and suggestions on the manuscript. Our responses (in blue) and the corresponding edits in the manuscript (in red) are shown below.

Reviewer #1 (Remarks to the Author):

Li et al: Enhancing atmospheric methane removal through Chlorine mediated chemistry-climate interactions.

General Comments

This is an important paper that reports some extremely interesting results. The research is very innovative and appears to be competently done with appropriate methodology (though I can't check it). The paper addresses a very important problem that is major topic of public debate. The findings will attract a great deal of interest both from the media and from policy makers.

I recommend publication after some revision.

Response 1:

We thank the reviewer for appreciating our efforts and for the constructive comments. We have carefully revised the manuscript accordingly. Please refer to the following response for details.

Background

There is a strong lobby, especially in the US and Europe, to 'geo-engineer' our way out of a significant part of global warming, by spraying salts into the troposphere (e.g. Oeste et al. Earth Syst. Dynam., 8, 1–54, 2017). Covering up the methane problem would then allow the fossil fuel jamboree to continue unimpeded, and hence this is an idea much liked by many investors and with significant political traction, not just in the US but also engaging oil and coal producing interests in many countries

Atmospheric chemists have generally been intuitively very sceptical of these ideas, on the general principles of "If you are in a hole, stop digging"; "the cover-up is worse than the crime"; and "There was an Old Woman who swallowed a fly/horse/dead of course" . But to date this intuitive scepticism has not really been backed up by detailed modelling to challenge the geo-engineering ideas. That absence has tended to leave the debate open to the proponents of planetary-scale action.

Response 2:

The reviewer very nicely summarized the motivation of our study, which is to provide a first-order estimate of the potential impacts of deliberately increasing the tropospheric chlorine burden, before any actions are actually taken.

We would like to note that in this conceptual study, we do not link our modelling setup to any specific climate intervention technique (e.g., via spraying iron salts as pointed out by the reviewer, or from marine cloud brightening). Instead, we adopt a simple model setup to allow the emission of molecular chlorine (Cl_2) over the global oceanic surface. The motivation for this setup is given in Response 3. Our focus is (1) to quantify the required chlorine emissions necessary to significantly reduce the global methane level, (2) to calculate the impacts of the increased chlorine on climate-relevant species (methane, tropospheric ozone, stratospheric ozone, stratospheric water vapour, sulfate aerosol, etc.) and the associated radiative forcings and surface temperature changes, and (3) to assess the unintentional environmental effects such as stratospheric O_3 depletion due to the increased concentrations of stratospheric chlorine.

We have revised the manuscript to further clarify the focus of the paper:

Page 11, line 364: “In this conceptual study, we do not link our modelling setup to any specific climate intervention technique (e.g., via spraying iron salts or marine cloud brightening). Instead, we adopt a simple model setup to emit Cl_2 over the global oceanic surface and to quantify the global impacts of the increased chlorine burden on atmospheric composition and the climate system.”

Page 4, line 105: “This includes analysing the global change in atmospheric composition, both of the intended change to CH_4 and the unintended changes to other atmospheric constituents (mainly to OH, tropospheric O_3 , sulphate aerosol, stratospheric O_3 , and stratospheric water vapor), and determining the associated radiative forcing and surface temperature response to these changes. Additionally, we indicate the possible environmental impacts due to the addition of chlorine, including the impacts on air quality and ocean acidification. We identify several uncertainties in our modelling results. Finally, we propose an agenda for future research on this potential climate mitigation methodology.”

Findings

Li et al enter this debate by using the standard CESM 1.1 Earth system Model and throwing in a great deal of chlorine, as proposed by the geo-engineering lobby. Then Li et al have a good look at what happens. Their results are quite surprising and there are several counter-intuitive findings. Two important issues stand out.

1. Li et al show that throwing chlorine into the air does not make things better unless you throw in a really huge amount of Cl. The threshold is 130 Tg Cl/yr. That's 3000 Titanics, or the payload of 7 million C-130 Hercules flights. They show it's above this threshold that injecting the chlorine enables an important chemical loss pathway for methane. To cut the methane lifetime by 50% needs an injection of 1900 Tg.

Response 3:

Indeed, our modelling results suggested that increasing the tropospheric chlorine atom burden has a nonlinear effect on the global methane burden, and we have conducted in total 15 model simulations (using the CESM model with various increases in chlorine emissions) to see whether there is a threshold in the response of methane to increased chlorine emissions. As the reviewer mentioned, we identified a threshold of 90 Tg Cl/yr, which is about a factor of 3 increase in tropospheric chlorine atom burden compared to present-day level, before methane levels decrease.

In our modelling simulations, the emissions of additional chlorine are from the global oceanic surface, instead of the free troposphere (e.g., via airplanes). Considering that we do not link our modelling results to specific chlorine climate intervention technique in this conceptual study, we do not discuss the details of emission setup, energy cost, associated CO₂ emissions, etc.; instead, the emission scenarios consider chlorine emissions from the Earth's ocean surface for the following reasons:

- (1) The potential emission of chlorine from Earth's surface is more feasible than emitting in the free troposphere which would require the use of aircraft.
- (2) Surface-based emissions reduces the energy cost of emitting chlorine.
- (3) Emitting from the surface could potentially make full use of sea-salt aerosol, a natural chloride-containing atmospheric species prevalent in the marine boundary layer.
- (4) Emissions over the ocean are safer than over land as they reduce the potential harmful effects on humans.
- (5) Emissions over the ocean surface will likely reduce the amount of chlorine transported to the stratosphere, compared to emissions in the free troposphere.

We have revised our description of the model setup to the following to further clarify our specification of chlorine emissions:

Page 11, line 368: “We have taken the following considerations into account when assuming the additional chlorine is emitted over the ocean surface, instead of in the free troposphere or over land: (1) to allow a feasible emission method which does not require aircraft; (2) to reduce the energy cost and associated CO₂ emissions required to emit chlorine; (3) to make full use of sea-salt aerosol, a natural chloride-containing atmospheric species prevalent in the marine boundary layer; (4) to reduce the potential harmful effects on humans (over land); (5) to reduce the injected amount of chlorine to the stratosphere.”

2. Li et al also show that chlorine emissions have a major impact on ozone, and injection of so much chlorine has a double effect on global warming – it decreases the warming impact of ozone as much as the impact of methane. So there's a big benefit here. assuming it doesn't matter that the acidification will cause a large-scale rearrangement of atmospheric chemistry and indeed the chemistry of the top of the ocean where the chlorine eventually rains out.

Response 4:

In this first global modelling assessment study of increasing the atmospheric chlorine burden, we try to be neutral about the socio-political implications of our modelling results, by avoiding referring to the effects as “beneficial” or otherwise. Instead, we attempt to quantify the possible outcomes, including the intended impact on methane, but also the unintended impacts on tropospheric ozone, tropospheric HCl, stratospheric ozone, etc. We have also identified some potentially important aspects that require further in-depth investigations.

We have revised the last paragraph of the introduction in the revised manuscript to further clarify the focus of the paper. Please refer to our response 2 for details.

Comments.

I am not able to check the modelling and methodology for mistakes. The CESM model is appropriate and the sensitivity simulation modelling appears to be done with standard methods. The 60 yr methane spin up seems reasonable. The text is well-written and clear, and the findings are presented in a lucid way.

I have one point of detail; the term "lifetime" is used for the e-folding lifetime (around 12 yr). That's quite normal in the modelling community (and indeed in IPCC) but it is not the 'lifetime' that is standard in the methane measurement community and in most public communications. For the non-modeller side of the methane debate, the "lifetime" is simply the geochemical lifetime - the burden divided by the flux – the replacement lifetime. That's about 9.2 years (e.g. Lan et al. 2021, *Phil Trans Roy Soc Lond A* 379: 20200440 – see their eqn 2.3. As this paper is going to be read widely outside the modelling community, I'd suggest using that version of 'lifetime'.

Response 5:

We thank the reviewer for pointing out the difference in calculating the lifetime in different research fields as well as for the suggestion to increase the readability in larger communities.

However, as the reviewer mentioned, in the modelling community and in the IPCC, as well as in our previous work (e.g., Li et al., *Nat. Commun.* 2022), “lifetime” is calculated in the same manner as in our current paper. These two methods both provide methane lifetime estimates of ~10 years, so choosing either one of the two methods is suitable. To maintain consistence within the modelling community, the IPCC, and our previous works, we prefer to keep the calculation method as it is.

We have added the following sentence in the revised manuscript to clarify our calculation method.

Caption of Fig. 1: “The CH₄ lifetime is defined as the chemical lifetime of CH₄ (obtained via dividing the atmospheric CH₄ burden by the CH₄ chemical loss rate).”

Now for several questions, which the authors should consider.

1. In the later sections of the paper (environmental policy) and in discussion of the global warming impact of removing the methane, the authors make no mention of the energy costs of the action of adding billions of tonnes of chlorine. Yet they are seriously considering injecting between 950 and 2840 Tg of Cl a year – that is roughly between 1 and 3 billion tonnes of chlorine, more if it is wafted up as iron chloride. To put that into context, pre-covid there were about 40 million aircraft take-offs in 2019, or less than 20 million in 2020. Assume each plane is an 80 tonne Boeing 737, then 40 million x 80 = somewhat over 3 billion tonnes. In other words, the geoengineers are proposing to inject annually into the troposphere a mass of the iron chloride equal to the total mass of aircraft that fly annually. That mass lifting of chlorine will have an energy cost – in effect a carbon dioxide cost – equal to the whole modern airline industry's emissions. So maybe that cost needs to be discussed here.

Response 6:

The reviewer raised a very important concern, the collateral energy cost and CO₂ emissions, of injecting chlorine. We completely agree with the reviewer that this aspect is important.

In our study, we emphasized the impacts of increasing the tropospheric chlorine emissions, and we quantified the potential environmental effects of this increased atmospheric chlorine burden. We did not link our work with any specific method (ship-based, ocean surface floating instrument, or plane-based). Therefore, our estimated environmental effects are in principle methodology independent. The energy cost or CO₂ emission may also vary significantly between different deployment technologies.

We agree with the reviewer that producing, transporting, and injecting a massive amount of chlorine into the atmosphere may have a substantial energy cost and carbon footprint. Any actions in the real atmosphere should consider both the environmental effects (that are quantified in our study) and the collateral cost and emissions.

We have added the following text in the section “Future research agenda” in the revised manuscript:

Page 10, line 329: “What is the environmental footprint (e.g., energy cost, CO₂ emissions, etc.) of the production, transportation, and deployment stage necessary to increase the atmospheric chlorine burden? ”

2. Perhaps there should also be a brief mention of the CO₂ emission costs of mining, pulverizing, and transporting several billion tonnes of iron chloride.

Response 7:

We thank the reviewer for the suggestion. Please see our response above.

3. The impact on tropospheric ozone is one of the most interesting findings in this paper, but perhaps there also should be a discussion of implications in the model for the Brewer-Dobson circulation. Much of the chlorine will have to be injected around the inter-tropical convergence zone, to allow the light and [OH] abundance to bite on the methane. But that inevitably means a lot of the chlorine will be injected up into the tropical stratosphere and the chloring will then travel round the Brewer-Dobson loop to the Polar Vortex and it will do a lot of interesting things on the way.....

Response 8:

We agree with the reviewer that a significantly increased tropospheric chlorine burden will inevitably enhance the injection of chlorine from the troposphere into the stratosphere, resulting in stratospheric ozone depletion.

Indeed, in the last paragraph of the section “Potential environmental impacts” in our original manuscript, we discussed the estimated effects on stratospheric ozone (particularly in the Antarctica) under different scenarios. Our results showed that significantly reducing the atmospheric methane burden (S1990 case) has a side effect of inducing a 5% decrease in the total stratospheric ozone and 17% decrease in the ozone burden within the Antarctica ozone hole area.

In light of these significant effects on stratospheric ozone, we have further elaborated this paragraph and brought the associated figures (Fig. S15 and S16 in the original manuscript) to the main text (Fig. 5 in the revised manuscript).

Page 8, line 253: “The enhancement in tropospheric chlorine atom also increases the stratosphere burden of chlorine (as much as a 44.9% increase in S1880 scenario average from 2020 to 2050) and even more in the Antarctic region (up to 242.6% increase; Fig. 5a). The significant increase in stratospheric chlorine results in enhanced depletion of stratospheric O₃ (R3) (1.0, 2.4, and 4.1% lower stratospheric O₃ in S630, S1250, and S1880, respectively, compared to RCP8.5, averaged from 2020 to 2050; Fig. 5b) which counter-acts the current stratospheric ozone recovery due to the phase-out of long-lived CFCs, HCFCs and Halons following the implementation of the Montreal Protocol. This is most important for the future evolution of the Antarctic O₃ hole during September and October, where we found 4.7, 18.3, and 37.4% lower stratospheric O₃ in S630, S1250, and S1880, respectively, compared to RCP8.5 (Fig. 5b).”

Fig. 5. Stratospheric chlorine and O₃ burden in the year 2050 in RCP8.5, S10, S630, S1250, S1880 scenarios. **(a)** Chlorine burden in the stratosphere and that in the Antarctic stratosphere (90°S to 70°S; 70hPa to 200hPa) during September and October. **(b)** O₃ burden in the in the stratosphere and that in the Antarctic stratosphere (90°S to 70°S; 70hPa to 200hPa) during September and October. The numbers on top of the sensitivity scenarios bars indicate the changes of chlorine/O₃ in these scenarios compared to that in RCP8.5 case. Note that both Y-axes in both panels start at values larger than zero.

Conclusion

To conclude: this paper is remarkably interesting and potentially important. The paper is clearly only scratching the surface of a very big puzzle, but it's a good start. While it is unlikely the geo-engineers will get their way, it's nevertheless possible they will indeed try something – it only takes one billionaire - and thus it is a problem that deserves attention. I strongly recommend the paper should be published – but after some revision.

Response 9:

We thank the reviewer for recognizing the importance and novelty of our work. Indeed, methane removal is a big topic that involves many scientific fields and stakeholders. While the idea of intentionally increasing tropospheric chlorine burden (in an attempt to increase the loss of methane through $\text{Cl}+\text{CH}_4$ reaction) has been mentioned in previous studies, the effects of increased amount of chlorine on atmospheric composition in general, climate, and the Earth system has not been quantified with a coherent and coupled modelling structure. Considering the non-linearity of the $\text{Cl}-\text{O}_3-\text{OH}-\text{CH}_4$ system, a thorough investigation of potential outcomes of changing the atmospheric chemistry is direly needed. We have followed the reviewer's suggestions and comments and revised the manuscript to better reflect our intentions and findings.

In the revision stage, in addition to addressing the comments raised by the above reviewers, we have further made two additional changes: (1) we have included a paragraph on the potential impacts of increased tropospheric chlorine burden on ocean acidity, which was contributed by Prof. Scott Doney from the University of Virginia, who is now added as a co-author, and (2) we have corrected a minor error in calculating the globally integrated chlorine emission (by not considering the fraction of ocean in our area weighting). In the revised manuscript, the corrected emission fluxes are slightly lower than those in the previously submitted version. Please note that all conclusions in the previously submitted version remain unchanged.

Response to Reviewer #2:

We thank the reviewer for the comments and suggestions on the manuscript. Our responses (in blue) and the corresponding edits in the manuscript (in red) are shown below.

Reviewer #2 (Remarks to the Author):

Li et al. investigated the effects of mitigating methane by emitting additional chlorine. They conducted sensitivity simulations in CESM and analyzed the influences of adding molecular chlorine on methane, ozone, their radiative forcing, and other environmental impacts. I totally agree with the author that “the potential environmental impacts on air quality and ocean acidity, must be carefully considered before any action is taken”. Climate issue is definitely important, but it’s tricky that whether it is the time to need some last resort to kill methane. Considering well-known harmful impacts of chlorine such as toxicity and acidity, proposing to emit large amount of reactive chlorine, in my opinion, is not conveying very good information.

The manuscript is interesting and well written, but I didn't find the results to be sufficiently novel, clear, and important for a high impactful journal. Most important, the authors added chlorine emissions in their simulations as molecular chlorine, but didn't clarify what chemical form of chlorine they proposed to add. Maybe one of the reactive chlorine gases (Cl₂, HOCl, etc.)? Due to different chemistry and depositions, depending on what species is going to emit, the analysis and conclusions in the manuscript could be different.

Response 1:

We thank the reviewer for the comments. We would like to clarify that in this paper, we are not “proposing to emit large amounts of reactive chlorine”. Instead, our intention is to quantify the intended and unintended impacts of emitting additional chlorine to the atmosphere, and estimate the amount of chlorine necessary to significantly reduce the methane burden and radiative forcing. As the reviewer also noticed, we stated that “*the potential environmental impacts on air quality and ocean acidity, must be carefully considered before any action is taken*” in the original manuscript.

In order to further clarify our intention, we have changed our title to be:

“Global environment implications of atmospheric methane removal through chlorine-mediated chemistry-climate interactions”.

The emitted chlorine species in our simulations is Cl₂. We have clarified it in the revised manuscript:

Page 4, line 96: “We conduct twelve sensitivity simulations from the present-day (2020) to mid-century (2050) in which molecular chlorine (Cl₂) is emitted at a constant rate over all ocean surfaces”.

In this conceptual work, we do not link our modelling setup to any specific climate intervention technique; instead, we adopt a simple model setup to allow the emission of molecular chlorine (Cl_2) over the global oceanic surface. Our intention is to provide a first-order estimate of (1) the chlorine amount necessary to significantly reduce the methane lifetime and burden, (2) the impacts of increased chlorine on climate-relevant species (methane, tropospheric ozone, stratospheric ozone, stratospheric water vapour, sulfate aerosol, etc.) and the associated radiative forcings and surface temperature changes, as well as (3) the potential environmental impacts of deliberately increasing the tropospheric chlorine burden.

The idea of increasing chlorine emission to reduce methane burden and lifetime in the atmosphere is increasingly discussed as an intervention approach to mitigate climate change. However, it is important to consider the potential environmental, social, and ethical implications of this approach through a holistic consequent analysis before an implementation occurs. The analysis should consider the potential impacts on air and water quality, ecosystem health, and human health, as well as the potential unintended consequences, such as changes in precipitation patterns or unintended shifts in ocean currents. Further analysis should then consider the potential ethical and social implications of this approach, including questions around ownership of the technology and its deployment, and potential impacts on marginalized communities. Given the early stage of this technology, it is critical that a comprehensive impact analysis be conducted. Our study is the first one to provide such an analysis, which is important and necessary to identify any unexpected outcomes, before any actions are taken in the real atmosphere.

As to the uncertainty of our study, we agree with the reviewer that different emitted species could lead to different conclusions in terms of the amount of chlorine required to significantly reduce the atmospheric methane lifetime and burden. In fact, there are other possible uncertainties that could result in different required chlorine emission/burden required to reduce methane. We have summarized these uncertainties and added a few paragraphs in the revised manuscript:

Page 8, line 282: “In this study, we have examined the potential impact of the proposed approach on a global scale. However, it should be noted that the response to this approach will vary regionally depending on the concentrations of ozone, NO_x , and other pollutants. Despite chlorine emissions being confined to the ocean surface, chlorine may still reach inland areas, resulting in different environmental implications compared to marine environments. Although these effects are accounted for in the model, they may be masked by the use of global averages.

It is important to note that the pH of aerosols is not specifically calculated in CESM. The effect on the pH of aerosols of any of the proposed methods of increasing the reactive chlorine burden in the atmosphere, whether through directly emitting reactive chlorine, adding a substance that activates chlorine in aerosols, producing new SSA from the ocean, or any other method, is not clear. Pye et al.²⁸ review the current state of atmospheric acidity and found that even drastic changes in sulfur dioxide and nitrogen oxide emissions across the US and Canada did not have

a proportional effect on aerosol pH; on the other hand, clouds and fog exhibit a higher sensitivity to such changes. Further investigation is necessary for more accurate analysis of the acidity change resulting from additional chlorine emissions.

The method used to inject chlorine will also affect the radiative forcing (RF). For example, if an additional substance is used to react with the chlorine in the aerosol and release it to gas phase, that substance will have different optical properties that will result in a different RF response and change the surface temperature response.

Our modelling results show the tipping point of the CH₄ response to the addition of chlorine for the RCP8.5 based scenarios occurs at 90 Tg Cl/year. However, this threshold will largely depend on O₃ and CH₄ concentrations and other atmospheric conditions. Therefore, the threshold should not be considered as an absolute number but a general tipping point that has to be calculated for specific atmospheric conditions and chlorine injection methods.

Overall, these factors should be explored in future studies to better understand the potential impact of this approach on a regional scale and refine our understanding of its effectiveness to allow policy makers to conduct a cost-effectiveness analysis before any implementation is considered.”

We have also added a few sentences regarding the oceanic acidity.

Page 7, line 238: “The modeled deposition of chemical acidity to the surface ocean in the form of HCl from reaction R2 is potentially an order of magnitude larger than the acidity flux caused currently by anthropogenic reactive sulfur and nitrogen. Assuming that the majority of the injected Cl₂ is eventually removed as HCl deposition into the ocean, the three model scenarios result in upper bounds of 18, 35, and 53 Tmol Cl/year for RCP8.5 S630, S1250, and S1880, respectively. This is substantially larger than the estimated anthropogenic global ocean deposition of reactive sulfur (0.8 Tmol S/year) and reactive nitrogen (2.7 Tmol N/year) from fossil fuel combustion and agriculture.²⁵ While not as large as the acidification caused by ocean uptake of anthropogenic CO₂, reactive sulfur and nitrogen fluxes are thought to exacerbate regional acidification in coastal waters downwind of anthropogenic sources. The effects of Cl emissions on surface ocean acid-base chemistry will depend on the specifics of the Cl₂ injection process. For example, an injection method that uses iron to release Cl₂ through a catalytic cycle²⁶ would impact HCl in the atmosphere differently. However, the possibility of a separation in the NaOH and HCl could still generate substantial changes to surface ocean acid-base chemistry.²⁷”

Specific comments:

- p3, end of first paragraph: both here and in abstract, the authors mentioned the addition of chlorine to the atmosphere has been proposed to mitigate global warming. However, the cited refs (10-12) here are more likely to be the traditional geoengineering. They emitted salt containing chloride, which is different from emitting molecular (reactive) chlorine. Could you please list some refs who have proposed adding reactive chlorine like you did here?

Response 2:

Thank you for this comment, the citations used here suggest adding iron containing aerosols in the atmosphere that may initiate a catalytic activation cycle, forming highly reactive chlorine radicals in the presence of sunlight. This cycle starts when aerosols containing iron uptake chlorine and produces iron chloride species. These iron chloride species absorb sunlight and produce a chlorine atom that combine and lead to the degassing of chlorine molecules (Cl_2). In the gas phase Cl_2 produce a reactive chlorine atom that oxidizes methane to produce carbon dioxide, water and hydrochloric acid. The hydrochloric acid may be reabsorbed by the aerosol, as it is highly soluble, renewing this cycle.

However, in our study, we do not link our model setup to any specific climate intervention technique. Instead, we emit the molecular chlorine (Cl_2) directly from oceanic surface in the model setup and attempt to quantify the global impacts of increased chlorine burden on atmospheric composition and the climate system. We have clarified this in the revised manuscript:

Page 11, line 364: “In this conceptual study, we do not link our modelling setup to any specific climate intervention technique (e.g., via spraying iron salts or marine cloud brightening). Instead, we adopt a simple model setup to emit Cl_2 over the global oceanic surface and to quantify the global impacts of the increased chlorine burden on atmospheric composition and the climate system.”

We have also revised the following sentence in the revised manuscript:

Page 3, line 71: “Some studies proposed intentionally adding chlorine to the atmosphere to decrease CH_4 concentration through iron salt aerosol.^{10,11} Horowitz, et al.¹² have shown that an increase in global methane occurs as the result of unintentional chlorine additions. Their main focus was to study marine cloud brightening as a means of reducing radiative forcing through generation of additional sea salt aerosol (SSA). However, all these studies (including intended and unintended chlorine additions) did not quantify the amount of chlorine that is needed to achieve a reduction in global methane.”

- In addition, some parts of refs (10-12) did quantify the possible consequences. The related statement is not true.

Response 3:

We have revised the relevant sentence as shown in the previous response.

- p7, 2nd paragraph: the threshold of 3.4×10^4 ppt looks too high, is this a typo? Furthermore, the toxin is related to chlorine gas (Cl_2), not Cl atom. Reactive chlorine gases have much larger concentrations than Cl.

Response 4:

The cited reference indeed refers to the ambient Cl₂ standard to be 0.034 ppm, which is 3.4x10⁴ pptv or 34 ppbv, “the WHO Task Group proposed that ambient levels of chlorine be about 0.034 ppm (0.1 mg m⁻³) to “protect the general population from sensory irritation,” and “significant reduction in ventilatory capacity”

We have replaced the figure of Cl with that of Cl₂ in the revised manuscript.

Page 7, line 228: “Although molecular chlorine is a known toxin at levels greater than 34 ppbv, globally averaged surface Cl₂ mixing ratios only reach up to 2 ppbv in S1880 scenario (Fig. S11c).”

Supplementary Figure S1. Global surface mixing ratios of (a) O₃, (b) NO_x (log scaled), and (c) Cl₂ in the year 2050 for RCP8.5, RCP6.0 and mitigation scenarios. In the mitigation scenario (S1250) based on RCP6.0, the surface mixing ratios are similar to those in RCP8.5 S1880.

- p5, last paragraph and fig.3: when saying sulfate aerosol, is it mean sulfate only or sulfate-ammonium-nitrate aerosol? Won't the chlorine induced large pH change affect the partitioning of this total inorganic aerosol?

Response 5:

We thank the reviewer for raising the question of aerosol pH changes induced by the increased chlorine burden.

Please note that in the original manuscript, by saying sulfate aerosol, we meant sulfate aerosol only.

We have added a paragraph on aerosol pH as an uncertainty in the revised manuscript. Please refer to Response 1 for the details.

In the revision stage, in addition to addressing the comments raised by the above reviewers, we have further made two additional changes: (1) we have included a paragraph on the potential impacts of increased tropospheric chlorine burden on ocean acidity, which was contributed by Prof. Scott Doney from the University of Virginia, who is now added as a co-author, and (2) we have corrected a minor error in calculating the globally integrated chlorine emission (by not considering the fraction of ocean in our area weighting). In the revised manuscript, the corrected emission fluxes are slightly lower than those in the previously submitted version. Please note that all conclusions in the previously submitted version remain unchanged.

Response to Reviewer #3:

We thank the reviewer for the comments and suggestions on the manuscript. Our responses (in blue) and the corresponding edits in the manuscript (in red) are shown below.

Reviewer #3 (Remarks to the Author):

This paper discusses the impact of future increases in tropospheric chlorine on methane lifetime and its climate impact. This discussion is framed in the context of intentional additional Cl emissions as a method to mitigate surface warming and covers scenarios with very large increases.

The author team are experts in chemistry-climate modelling and the effects of halogens. In this work they have used a state-of-the-art chemistry-climate model.

The paper contains some interesting model sensitivity experiments but, in my opinion, it is not suitable for publication in a form like this in a journal like Nature Communications. The feedbacks of Cl on CH₄ lifetime have already been discussed in recent studies cited, so the motivation for this paper in such a high-level forum seems exaggerated.

Response 1:

We thank the reviewer for recognizing our work of interest.

We agree with the reviewer that the impact of current levels of chlorine (as well as forecasted levels based on probable societal-economic development and climate change) on atmospheric methane lifetime has been discussed in recent studies. However, the present work goes beyond the content of the existing literature. Here we highlight three new findings from our study:

- (1) The atmospheric and climate system is a highly nonlinear system. A small perturbation in one atmospheric composition leads to nonlinear response in the entire system. In fact, our modelling results suggest that if we keep increasing the tropospheric chlorine burden, the atmospheric methane burden first increases then decreases. This has not been reported before.
- (2) As Reviewer #1 mentioned, there is an active group advocating the idea of deliberately increasing the tropospheric chlorine burden to reduce the global methane burden and radiative forcing. Our study enters this public discussion and quantifies the main potential intended and unintended effects of increased tropospheric chlorine burden on the atmospheric, climate, and Earth system, including the impacts on global methane, tropospheric ozone, stratospheric ozone, tropospheric HCl, etc. The quantification of such effects have not been reported before.
- (3) In addition to the effects of increased chlorine burden on atmospheric compositions, our study also calculates the impacts on the radiative forcings not only for methane, but also for other key affected short-lived climate forcers (ozone, stratospheric water vapour, and sulfate aerosol) and the associated surface temperature changes. Our study is the first to investigate these impacts.

We have revised the manuscript to highlight the novelty of our results:

Page 3, line 71: “Some studies proposed intentionally adding chlorine to the atmosphere to decrease CH₄ concentration through iron salt aerosol.^{10,11} Horowitz, et al.,¹² have shown that an increase in global methane occurs as the result of unintentional chlorine additions. Their main focus was to study marine cloud brightening as a means of reducing radiative forcing through generation of additional sea salt aerosol (SSA). However, all these studies (including intended and unintended chlorine additions) did not quantify the amount of chlorine that is needed to achieve a reduction in global methane.”

Page 3, line 87: “However, the potential impacts of a significantly larger atmospheric chlorine burden on atmospheric composition, radiative forcing, and surface temperature remain unexplored.”

Page 4, line 101: “This paper quantifies the globally averaged impact of additional chlorine emissions as a potential climate intervention technique. A homogeneous addition of chlorine species over all ocean surfaces may not be feasible in this respect but is chosen here as a pragmatic starting point. We consider a multitude of cases with different chlorine emissions but omit regional analysis to show a synthesis of global impacts on the atmospheric chemistry and climate. This includes analysing the global change in atmospheric composition, both of the intended change to CH₄ and the unintended changes to other atmospheric constituents (mainly to OH, tropospheric O₃, sulphate aerosol, stratospheric O₃, and stratospheric water vapor), and determining the associated radiative forcing and surface temperature response to these changes. Additionally, we indicate the possible environmental impacts due to the addition of chlorine, including the impacts on air quality and ocean acidification. We identify several uncertainties in our modelling results. Finally, we propose an agenda for future research on this potential climate mitigation methodology.”

My main points are summarised below.

1) Recent papers, e.g. ref #11 and the group’s own Nat Comms paper #16, have already noted the feedback on Cl emissions on the CH₄ lifetime. We know that the naïve suggestion from some geoengineering papers on scaling the current impact of Cl on CH₄ is wrong. Therefore, I do not think that the motivation is there for another Nat Comms paper.

Response 2:

We thank the reviewer for the comment about the motivation/novelty of this study. Please refer to our response above for the new findings from the present work.

As for the impacts of increased chlorine on methane, we agree with the reviewer’s comment, “scaling the current impact of Cl on CH₄ is wrong”. However, we believe that we need to provide a quantified estimate to support our point of view. The present work is driven by this motivation.

We have adopted a state-of-the-art Earth system model with comprehensive halogen chemistry, conducted 15 sensitivity simulations with various increases in tropospheric chlorine burden, quantified the impacts on global climate-relevant species (methane, tropospheric ozone, stratospheric ozone, stratospheric water vapour, sulfate aerosol, etc.), calculated the impacts on radiative forcing and the associated surface temperature, and identified several key questions that are needed in future studies. By quantifying many impacts of increased chlorine on the atmospheric system within a coherent framework (modelling system), we are able to support our conclusion that a massive increase in tropospheric chlorine burden is needed to significantly reduce the global methane burden, inevitably leading to many unintended effects that require extra attention.

2) The paper seems to fall between the aim of a high level discussion (in the main text) and providing the quantitative model detail (in the Supplementary Information). With so many references to the 16 supplementary figures it is difficult to read if one wants to understand the details and limitations of the model. I am left with the feeling that the authors should have better written a longer paper in a subject-specific journal where the model results could be presented clearly. I note that the motivating references #10, #11 and #12 are all journals that allow more detailed discussion.

Response 3:

It is a common practice to show the key results as figures/tables in the main text and detailed supporting results in the supplement. Although there are 16 figures in the supplement, these figures are detailed results for the readers who are interested in relevant information. We believe that referring to these supplementary figures do not interfere with the understanding of the main story. Please note that these supplement figures are also published online together with the main text.

In light of the significant effects of the increased chlorine burden on the environment, we have moved the supplementary figures S15 and S16 (the impacts on stratospheric ozone) to the main text in the revised manuscript, and modified the text as follows.

Page 8, line 253: “The enhancement in tropospheric chlorine atom also increases the stratosphere burden of chlorine (as much as a 44.9% increase in S1880 scenario average from 2020 to 2050) and even more in the Antarctic region (up to 242.6% increase; Fig. 5a). The significant increase in stratospheric chlorine results in enhanced depletion of stratospheric O₃ (R3) (1.0, 2.4, and 4.1% lower stratospheric O₃ in S630, S1250, and S1880, respectively, compared to RCP8.5, averaged from 2020 to 2050; Fig. 5b) which counter-acts the current stratospheric ozone recovery due to the phase-out of long-lived CFCs, HCFCs and Halons following the implementation of the Montreal Protocol. This is most important for the future evolution of the Antarctic O₃ hole during September and October, where we found 4.7, 18.3, and 37.4% lower stratospheric O₃ in S630, S1250, and S1880, respectively, compared to RCP8.5 (Fig. 5b).”

Fig. 5. Stratospheric chlorine and O₃ burden in the year 2050 in RCP8.5, S10, S660, S1330, and S1990 scenarios. **(a)** Chlorine burden in the stratosphere and that in the Antarctic stratosphere (90°S to 70°S; 70hPa to 200hPa) during September and October. **(b)** O₃ burden in the in the stratosphere and that in the Antarctic stratosphere (90°S to 70°S; 70hPa to 200hPa) during September and October. The numbers on top of the sensitivity scenarios bars indicate the changes of chlorine/O₃ in these scenarios compared to that in RCP8.5 case. Note that both Y-axes in both panels start at values larger than zero.

3) There must be considerable model uncertainties in the specific numbers quoted, but this is not acknowledged. The abstract gives some numbers but the method (i.e. running a complex model) is not stated. To the non-expert I think the numbers are presented with more confidence than they can really be known (e.g. the 130 Tg/yr tipping point). Models can differ significantly in their O₃ burden, halogen burden and so on. This paper presents some results in too absolute a way.

Response 4:

We agree with the reviewer that model uncertainties should be mentioned and acknowledged in the manuscript.

In response to the reviewer's comment, we have added a new section "Uncertainty analysis" in the revised manuscript to identify the uncertainties associated to the reported results.

Page 8, line 282: "In this study, we have examined the potential impact of the proposed approach on a global scale. However, it should be noted that the response to this approach will vary regionally depending on the concentrations of ozone, NO_x, and other pollutants. Despite chlorine emissions being confined to the ocean surface, chlorine may still reach inland areas, resulting in different environmental implications compared to marine environments. Although these effects are accounted for in the model, they may be masked by the use of global averages.

It is important to note that the pH of aerosols is not specifically calculated in CESM. The effect on the pH of aerosols of any of the proposed methods of increasing the reactive chlorine burden in the atmosphere, whether through directly emitting reactive chlorine, adding a substance that activates chlorine in aerosols, producing new SSA from the ocean, or any other method, is not clear. Pye et al.²⁸ review the current state of atmospheric acidity and found that even drastic changes in sulfur dioxide and nitrogen oxide emissions across the US and Canada did not have a proportional effect on aerosol pH; on the other hand, clouds and fog exhibit a higher sensitivity to such changes. Further investigation is necessary for more accurate analysis of the acidity change resulting from additional chlorine emissions.

The method used to inject chlorine will also affect the radiative forcing (RF). For example, if an additional substance is used to react with the chlorine in the aerosol and release it to gas phase, that substance will have different optical properties that will result in a different RF response and change the surface temperature response.

Our modelling results show the tipping point of the CH₄ response to the addition of chlorine for the RCP8.5 based scenarios occurs at 90 Tg Cl/year. However, this threshold will largely depend on O₃ and CH₄ concentrations and other atmospheric conditions. Therefore, the threshold should not be considered as an absolute number but a general tipping point that has to be calculated for specific atmospheric conditions and chlorine injection methods.

Overall, these factors should be explored in future studies to better understand the potential impact of this approach on a regional scale and refine our understanding of its effectiveness to

allow policy makers to conduct a cost-effectiveness analysis before any implementation is considered.”

We have also revised the manuscript to present the numbers and results in a less absolute way.

Abstract: “our modelling results suggest that”

Page 4, line 118: “Based on our CESM modelling results,”

Page 5, line 168: “The concentrations of CH₄ and O₃ in the base scenario will affect this threshold. Similarly, scenarios with more locally concentrated emissions of Cl may also change this threshold due to the non-linear chemistry involved.”

4) Following 3) I think an important issue for any research agenda would be multi-model intercomparisons to test the robustness and quantitative details of the feedbacks discussed.

Response 5:

We thank the reviewer for the suggestion to include multi-model intercomparison in the future research agenda. We have added this item in the revised manuscript:

Page 10, line 332: “Multi-model intercomparison studies should be conducted to investigate the impacts of various increased tropospheric chlorine burden on the atmospheric, climate, and Earth system.”

Please note that in the revised manuscript, we have added a paragraph in the section “Uncertainty analysis” to discuss the uncertainty in the threshold estimate. Please also see our Response 5 above.

In the revision stage, in addition to addressing the comments raised by the above reviewers, we have further made two additional changes: (1) we have included a paragraph on the potential impacts of increased tropospheric chlorine burden on ocean acidity, which was contributed by Prof. Scott Doney from the University of Virginia, who is now added as a co-author, and (2) we have corrected a minor error in calculating the globally integrated chlorine emission (by not considering the fraction of ocean in our area weighting). In the revised manuscript, the corrected emission fluxes are slightly lower than those in the previously submitted version. Please note that all conclusions in the previously submitted version remain unchanged.